# The Over-Certainty Phenomenon in Modern Test-Time Adaptation Algorithms

**Fin Amin**                                                                    *samin2@ncsu.edu*
*Department of Electrical and Computer Engineering*
*North Carolina State University*

**Jung-Eun Kim**[*]                                                             *jung-eun.kim@ncsu.edu*
*Department of Computer Science*
*North Carolina State University*

**Reviewed on OpenReview:** *https://openreview.net/pdf?id=AGQRij8iUC*

## Abstract

When neural networks are confronted with unfamiliar data that deviate from their training set, this signifies a domain shift. While these networks output predictions on their inputs, they typically fail to account for their level of familiarity with these novel observations. Prevailing works navigate test-time adaptation with the goal of curtailing model entropy, yet they unintentionally produce models that struggle with sub-optimal calibration—a dilemma we term the over-certainty phenomenon. This over-certainty in predictions can be particularly dangerous in the setting of domain shifts, as it may lead to misplaced trust. In this paper, we propose a solution that not only maintains accuracy but also addresses calibration by mitigating the over-certainty phenomenon. To do this, we introduce a certainty regularizer that dynamically adjusts pseudo-label confidence by accounting for both backbone entropy and logit norm. Our method achieves state-of-the-art performance in terms of Expected Calibration Error and Negative Log Likelihood, all while maintaining parity in accuracy.

## 1 Introduction

When encountering new environments, humans naturally adopt a cautious approach, assimilating the novelty to guide their decision-making. This inherent ability to assess unfamiliarity and adjust certainty has not been entirely emulated in artificial neural networks. Unlike humans who might exhibit hesitation in unknown situations, many test-time adaptation (TTA) algorithms lack an explicit mechanism to modulate certainty in response to the novelty or unfamiliarity of their inputs.

Deep learning has never been a stranger to the challenges of uncertainty. Over the past few years, the miscalibration problem of modern neural networks has gained substantial attention, as highlighted by works such as Guo et al. (2017), Abdar et al. (2021), Liang et al. (2017), and Pampari & Ermon (2020). However, how TTA algorithms themselves alter calibration is understudied. In this paper, we uncover the *over-certainty phenomenon*, a phenomenon that plagues many modern TTA algorithms by harming model calibration.

A prevailing strategy among TTA algorithms is the minimization of entropy, either as an explicit target or as an inherent by-product of their methodology. And while this might bolster accuracy metrics, our research indicates a concerning trend: excessive entropy reduction can be detrimental to model calibration. What makes this trend more problematic is that it occurs within the context of a new domain, where epistemic uncertainty should typically be greater. In our work, we measure certainty as the inverse of Shannon entropy of a model's output after a softmax operation: $H(f(x))^{-1} = (\texttt{Entropy}_2(\texttt{SoftMax}(f(x))))^{-1}$.

---

[*]Corresponding author

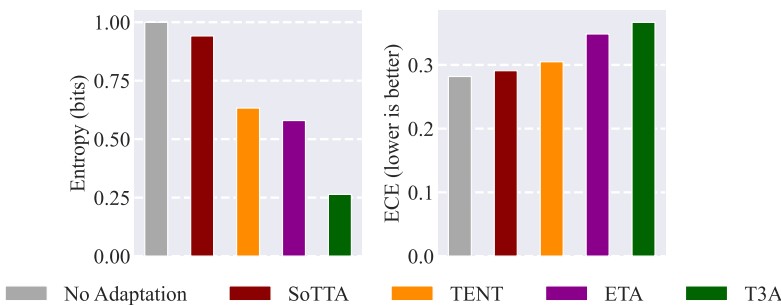

Figure 1: This chart shows four modern TTA algorithms adapting EfficientNet to the *clipart* domain. Minimizing entropy is a common objective in recent work. However, this can have consequences on model calibration.

To further frame our discussion, TTA is used when a model, trained on a source domain $(X_s, Y_s)$, is presented with the challenges of a different yet analogous target domain $(X_t)$ with no labels. In our problem setting, we do not assume access to $X_t$ until we adapt. The nuances between these domains, commonly termed as domain shift, can introduce significant disruptions in model performance. TTA, in its essence, aspires to adapt the insights harvested from the source domain and apply them proficiently to the target domain, bypassing the need for labeled data in the latter. TTA is difficult because models must generalize to unfamiliar distributions without access to labeled data or strong assumptions about how the target domain relates to the source Fang et al. (2022); David et al. (2010). Furthermore, as we will elaborate in the next section–it can be especially problematic to be too certain within the context of a domain shift.

With the purpose of addressing these intertwined challenges, we introduce *Dynamic Entropy Control*. This TTA technique seeks to augment accuracy and improve model calibration. By interweaving calibration into the core learning process, we produce a TTA adjustment algorithm that jointly improves accuracy while managing epistemic uncertainty. To summarize, our contributions are:

- The identification of the over-certainty phenomenon. We emphasize that our work is not about the well-studied tendency of models to become less calibrated under distribution shift Pampari & Ermon (2020); Ovadia et al. (2019). Instead, we show how a recent trend in the design of TTA algorithms can yield models that are even more miscalibrated than the unadapted baseline. To the best of our knowledge, no prior work—including Press et al. (2024)—explicitly analyzes this pattern while also providing thorough empirical evidence and mechanistic insight into why over-certainty emerges during adaptation.

- *Dynamic Entropy Control*, a new TTA algorithm that achieves SOTA calibration in all of the four datasets and SOTA accuracy uplifts in the majority of domain shifts.

Our study is scoped to test-time adaptation algorithms that follow the prevailing trend of minimizing entropy on unlabeled observations in the classification setting. While we acknowledge the existence of TTA methods that do not follow this paradigm (e.g., Liang et al. (2020); Tang et al. (2020)), they fall outside the focus of our analysis. **Unless otherwise stated, all mentions of "TTA algorithms" in this work refer specifically to entropy-reduction-based approaches.**

## 1.1 Why Care About Over-Certainty?

We emphasize that calibrating uncertainty estimates has ramifications that go well beyond quantitative metrics. For instance, in real-world scenarios that rely on trustworthy estimates of model confidence (e.g., autonomous driving, active learning, anomaly detection), an overconfident model can lead to harmful misjudgments You et al. (2022). As a concrete illustration, consider a self-driving car that uses a classifier to detect whether a neighboring lane is occupied:

- *Pre-adaptation (low certainty):* The vehicle's classifier might cautiously predict "safe to switch lanes" but at a low confidence. This low certainty prevents the car from making a risky maneuver.

- *Post-adaptation (artificially high certainty):* After applying a traditional TTA algorithm, the same classifier might become *overly certain* about its "safe to switch lanes" prediction. In reality, if this confidence is misplaced, the system can make a high-stakes error that jeopardizes safety.

Thus, our approach seeks to preserve or improve classification accuracy *and* produce calibrated uncertainty estimates, ensuring that decisions made based on confidence thresholds are more reliable. Such reliability is also valuable in other domains that hinge on calibrated outputs, such as fairness-sensitive applications Hébert-Johnson et al. (2018); Creager et al. (2021), where miscalibration can exacerbate biases or distort outcome distributions.

**Fairness Considerations.** As suggested in Pleiss et al. (2017), there is a growing recognition of how calibration can influence fairness. Overconfident models may disadvantage certain groups when high-confidence predictions guide resource allocation or risk assessments. Ensuring well-calibrated probabilities is therefore crucial but often overlooked in current TTA approaches.

## 2    Related Work

Our literature survey covers methodologies catered towards updating a neural network on unlabeled data. For the sake of brevity, we will refer to unlabeled data as "observations." This section gives an overview of the work done to improve networks on the fly. We start by introducing earlier work, such as dictionary learning techniques and lead our way into recent developments. We elaborate the algorithms we compare with in detail to give mechanistic insight on why over-certainty happens. The next section covers how calibration is measured and improved in the context of a domain shift. Lastly, we discuss assessing the reliability of an observation. Furthermore, we provide a supplementary discussion of related work in Appendix A.3.

### 2.1    The Test-Time Entropy Reduction Paradigm

The phrase "self-taught learning" was coined by Raina et al. (2007). In this work, the authors utilize observations to find an optimal sparse representation of said observations. This sparse representation is used to train their model in lieu of the ordinary training set to improve out-of-distribution (OOD) performance.

Work in this field has extended to a variety of approaches such as the use of pseudo-labeling to exploit the existing model's predictions as target labels Lee et al. (2013); Mancini et al. (2018); Wang et al. (2022). Pseudo-labeling can be thought of under the guise of knowledge distillation (KD) Hinton et al. (2015); Gou et al. (2021). KD is a transfer learning paradigm where a large neural network, known as the teacher, transfers "knowledge" to a smaller "student" network. Succinctly, the student is trained to match the output of the teacher when given the same input as the teacher Stanton et al. (2021). In pseudo-labeling, the teacher and student are the same network.

The TENT algorithm introduces the trend of **test-time entropy minimization** Wang et al. (2020a). Entropy minimization can be thought of as using pseudo labels with the cross entropy loss function. In other words, the entropy minimization works by using gradient descent to minimize:

$$L_{TENT} = -\sum_{y \in C} f(y|x) \log f(y|x) \tag{1}$$

to update the model's batch-normalization parameters.

More recent advancements in this trend include EATA/ETA Niu et al. (2022), T3A Iwasawa & Matsuo (2021) and SoTTA Gong et al. (2024). ETA[1] advances on TENT by making sure that observations are *reliable*

---

[1]The authors of EATA/ETA introduce two similar algorithms, for our paper, we focus on ETA which performs the best between the two.

and *non-redundant* before they are used for updating the batch-normalization parameters. To do this, they compute a sample adaptive weight, $\mathcal{S}(x)$, for each observation before minimizing entropy:

$$L_{ETA} = -\mathcal{S}(x) \sum_{y \in C} f(y|x) \log f(y|x) \tag{2}$$

where $\mathcal{S}(x)$ is a function of the entropy of the model towards the batch sample (i.e., the reliability) and the similarity to what it has seen before (i.e., non-redundancy). Similar to the aforementioned methods, SoTTA minimizes entropy via sharpness-aware-minimization (SAM) Foret et al. (2020). The algorithm employs high-confidence uniform sampling to create a memory bank of size $N_{SoTTA}$ which stores reliable and class-balanced observations. This is done by using confidence to asses if an observation should be used for adaptation. Confidence is defined as:

$$\mathcal{C}_f(x) = \max_{i=1,\dots,n} \frac{\exp(f(x)_i)}{\sum_{j=1}^{n} \exp(f(x)_j)} \tag{3}$$

If $\mathcal{C}_f(x) > \mathcal{C}_0$, where $\mathcal{C}_0$ is some pre-defined confidence threshold, then $x$ is saved into memory. Afterwards, the SAM optimizer is used to minimize entropy (equation 1) via two backpropagation steps. The T3A algorithm Iwasawa & Matsuo (2021) differs from the previous three as it focuses on updating the *prototypes* Snell et al. (2017) of each class during test time:

$$S_k^t = \begin{cases} S_k^{t-1} \cup \{f(x)\}, & \text{if } \hat{y} = y_k \\ S_k^{t-1}, & \text{else.} \end{cases} \tag{4}$$

$$c_k = \frac{1}{|S_k|} \sum_{z \in S_k} z \tag{5}$$

where $c_k$ represents the centroid of the prototypes of a class $k \in C$, where $C$ is the number of classes. We define the feature extractor, $\psi$, as all the layers of the backbone before the final dense layer. The final dense layer, $\phi$, is what we refer to as the classifier, it is composed of the class centroids. We denote the output of the feature extractor as $z = \psi(x)$.

Unlike TENT, SoTTA, or ETA, T3A does not explicitly reduce entropy as it does not use a loss function, however the authors claim that entropy reduction is an effect of using their algorithm. Similar to ETA, this algorithm filters less reliable samples during equation 4 by only keeping the $M$ lowest entropy prototypes for each class. Therefore, the algorithm stores $C \cdot M$ prototypes.

## 2.2 Neural Network Calibration

Neural network calibration has been of intense interest in recent years due to the critical role of confidence values, which reflect the probability assigned to predictions, in various applications. For instance, BranchyNet Teerapittayanon et al. (2017) uses neural network confidence to enable early exits for faster inference, relying on high confidence at intermediate layers. However, Zhu et al. (2022) highlights the prevalent issue of certainty calibration in deep networks, where models often display overconfidence or underconfidence, likely due to overfitting during training. Ovadia et al. (2019); Minderer et al. (2021) explore this concept further by measuring a model's Expected Calibration Error (ECE) and Negative Log-Likelihood (NLL). ECE measures how closely the confidence levels of a model's predictions match the actual probability of those predictions being correct. It calculates the average absolute difference between predicted confidence and the true outcome frequencies, providing a metric for the reliability of the model's probabilistic outputs. NLL, on the other hand, captures both the model's calibration and sharpness by quantifying how well the predicted probabilities align with the observed outcomes, penalizing overconfident yet incorrect predictions more heavily. We follow their convention and use ECE and NLL as our measures for calibration error. They notice models calibrated on the validation set tend to be well calibrated on the test set, but are not properly calibrated to shifted data.

Recent work has also investigated solutions to this phenomenon. Guo et al. (2017) discusses a technique known as temperature scaling while Wei et al. (2022) approaches this problem by regularizing the logit norm.

Table 1: Comparison of Shannon Entropy, ECE, and NLL across the Home Office dataset domains. Results use the MobileNet backbone. Our reduction in ECE and NLL is statistically significant ($p < 0.01$) while maintaining competitive accuracy.

| Algorithm | Shannon Entropy | | | | ECE (lower is better) | | | | NLL (lower is better) | | | |
|---|---|---|---|---|---|---|---|---|---|---|---|---|
| | Art | Clipart | Product | Real World | Art | Clipart | Product | Real World | Art | Clipart | Product | Real World |
| No Adapt | 0.950 | 0.951 | 0.704 | 0.689 | 0.302 | 0.316 | 0.183 | 0.169 | 3.196 | 3.330 | 1.777 | 1.638 |
| DEC (ours) | 2.239 | 2.615 | 2.081 | 1.795 | **0.080** | **0.047** | **0.039** | **0.018** | **2.077** | **2.127** | **1.328** | **1.212** |
| T3A | 0.188 | 0.231 | 0.138 | 0.170 | 0.439 | 0.418 | 0.236 | 0.248 | 7.904 | 7.756 | 3.536 | 3.321 |
| ETA | 0.940 | 0.966 | 0.735 | 0.678 | 0.313 | 0.324 | 0.208 | 0.172 | 3.180 | 3.438 | 1.966 | 1.768 |
| TENT | 0.918 | 0.895 | 0.653 | 0.636 | 0.299 | 0.308 | 0.187 | 0.172 | 3.120 | 3.234 | 1.780 | 1.632 |
| SoTTA | 0.977 | 0.971 | 0.701 | 0.685 | 0.279 | 0.301 | 0.186 | 0.167 | 2.938 | 3.240 | 1.825 | 1.632 |

More classical solutions to this problem exist as well; Zhang et al. (2021) and Müller et al. (2019) consider label smoothing to address this issue. Note that these techniques are addressed at calibrating the underlying backbone but have *not* been investigated with respect to TTA algorithms themselves.

### 2.3 Detecting Out of Distribution Data and Assessing Reliability

An increasing body of research examines how to assess whether and how closely a new observation aligns with a model's training distribution. For example, the authors of Tian et al. (2019) observe that if an autoencoder was trained to reconstruct inliers, it would have a greater reconstruction error when reconstructing OOD data. Schlegl et al. (2017) and Zenati et al. (2018) approach this issue by observing that the discriminator of a GAN learns whether or not a given input is an inlier. Many other works delve into this domain Bendale & Boult (2016); Ming et al. (2022); Park et al. (2023); Jiang et al. (2023); Wu et al. (2023); Miao et al. (2023); Fang et al. (2022). Regarding the TTA algorithms we compare against, the most common proxy for reliability is entropy on the observation.

## 3 What is the Over-Certainty Phenomenon?

In this work, we present evidence for what we dub the *over-certainty phenomenon* (OCP) of contemporary TTA approaches. This phenomenon is that TTA algorithms tend to miscalibrate their underlying backbone networks by causing their predictions to be excessively certain. Modern TTA algorithms often strive to decrease test-time entropy. However, as shown in Fig. 1, this entropy reduction may increase ECE and NLL because the models become overly certain on their predictions.

This phenomenon of existing algorithms causing models to become overly certain presents itself across many other datasets. A compelling example is given in Table 2 which agglomerates calibration errors over 15 domain shifts; in this table, we see a clear trend of entropy reduction (certainty increasing) and sub-optimal calibration. Another example is provided in Table 1, T3A reduces entropy by a factor of about 4 in the *art, clipart* and *product* domains. As before, it causes ECE to worsen compared to the baseline. We do not claim that TTA algorithms should *always* strive to increase backbone uncertainty; poor calibration can also be caused by under-certainty and there exist cases where reducing entropy compared to baseline improves calibration. However, we find that the resulting calibration is still sub-optimal. Despite these complexities, our investigation reveals a consistent pattern: *the over-certainty phenomenon causes sub-optimal model calibration*, a significant concern for safety, robustness, and reliability.

### 3.1 What Causes the Over-Certainty Phenomenon?

We identify two plausible causes of the OCP, the first issue is that modern TTA algorithms aim at minimizing backbone entropy too aggressively. In the case of TENT, ETA, and SoTTA, their loss functions, equation 1 and equation 2, explicitly aim at reducing a model's entropy. Regarding TENT, there is no regularization of this process. In the case of ETA, the algorithm uses a *reliability score*, $S(x)$, which aims at weighing observations differently but does not regularize the distributions of the pseudo-labels. Unlike the other two,

SoTTA minimizes entropy twice per iteration. The authors of T3A claim that entropy reduction is an effect of using their algorithm. In fact, they show in certain datasets T3A reduces entropy more than TENT does.

Another issue is how existing methods evaluate observation *reliability*, the suitability of a model's prediction for use for adaptation. Previous works, ETA and T3A, tap into the power of model certainty, using it to weigh the influence of observations. ETA assesses reliability by ensuring that observations meet a certain entropy threshold; similarly, T3A uses entropy to sort the importance of class prototypes. However, as prior work has shown, there are drawbacks in using entropy as a proxy for reliability in this manner Wei et al. (2022). To illustrate our point, we give a toy example of how using entropy can lead to a misleading conclusion:

**Example 1.** *Suppose that we analyze the classifier while classifying between two classes with class centroids, $c_0$ and $c_1$. This is done by taking the output of the feature extractor, $\psi(x) = z$, and computing the dot product between the centroids and $z$.*

$$g = [z \cdot c_0, z \cdot c_1] \tag{6}$$

*Consider $g_{t1}, g_{t2}$ and $g_s$ as vectors representing the inner products related to two observations, $x_{t1}$ and $x_{t2}$, and to a specific training sample, $x_s$. Specifically, $g$ corresponds to the dot products between the output of the feature extractor and the class centroids. As an example, let's assume:*

$$g_s = [8.0, 7.29]; \ g_{t1} = [1.92, 1.00]; \ g_{t2} = [6.10, 6.50];$$

*If we take the softmax of these vectors and compute the entropy, we get $\texttt{Entropy}_2(\texttt{SoftMax}(g))$ for $g_s, g_{t1}$ and $g_{t2}$, as 0.92 bits, 0.86 bits and 0.97 bits, respectively.*

Notice that if we consider the entropy of these three vectors as a proxy for reliability, we would consider $x_{t1}$ to be more reliable than $x_{t2}$, despite $x_{t2}$ having considerably greater inner product with the class centroids. It is highly likely that the values of $g_{t1}$ occurred due to spurious feature correlations between $x_{t1}$ and the class centroids. In fact, in the scenario above, $x_{t1}$ would be deemed to be more reliable than the genuine source domain observation $x_s$. Note that an analogous remark could be made on using confidence instead of entropy. Our analysis is not contrived; in Fig. 2, as we increase the domain shift intensity, the observation logit norm decays and has higher variance.

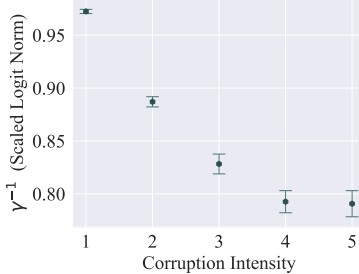

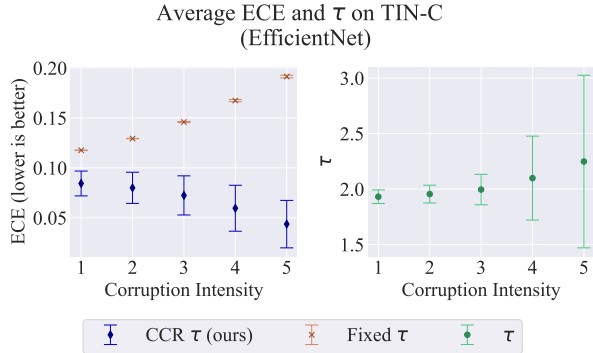

Figure 2: As domain shift intensity increases, the bottom quantile of the observation logit norm decreases. The $\gamma^{-1}$ value (step 8) from our `CCR` algorithm represents the ratio between the $l_2$ norms of the observation logits, $z$, and the training-set logits, $\kappa$. As $z$ decreases, $\gamma$ increases; this regularizes low-logit-norm observations more aggressively (step 9). Vertical bars indicate domain-to-domain standard deviations.

Figure 3: Our ablation study highlights the effectiveness of the `CCR` algorithm compared to a fixed $\tau$ optimized for minimal ECE on the source-domain training set. The rightmost figure displays the computed $\tau$ values, with vertical bars indicating domain-to-domain standard deviations.

Furthermore, existing approaches do not consider a model's certainty on the source domain. For example, ETA's reliability paradigm rejects observations which have $H \geq 0.4 \cdot \ln(C)$. What if we *expect* (i.e. there was

high entropy on the training set) the model to have high entropy? By only evaluating the target domain's certainty without juxtaposing it against the source domain certainty, there is a lack of reference in terms of assessing the reliability of the observation. We address this issue via the $h_0$ parameter explained in the next section.

# 4 The Dynamic Entropy Control TTA Algorithm

To ameliorate the over-certainty phenomenon, we introduce Dynamic Entropy Control (`DEC`) (Algorithm 2). DEC refines the model's certainty levels, aligning them more closely with its actual accuracy, by selectively adjusting the temperature parameter during the pseudo-labeling process (line 6). This is achieved without directly altering ground truth labels, instead focusing on the tempering of logits through temperature adjustments. Our approach can be implemented using a single model which alternates weights. To facilitate understanding, we explain our algorithm as if it uses two distinct models: the teacher and the student. The "teacher" model, $f_{te}$, is simply our backbone before any adaptation. The "student" model, $f_s$, is the model that is adapted.

---

**Algorithm 1** Compute Certainty Regularizer (CCR)

**Input**: $f_{te}(X), h_0, h_{max}, t_{min}, t_{max}, \kappa$
**Output**: $T_{vec}$
 1: $Z = f_{te}(X)$ {get logits}
 2: $H_{vec} = \texttt{Entropy}_2(\texttt{SoftMax}(Z))$ {entropy for each sample}
 3: $H_{diff} = H_{vec} - h_0$ {compare entropy with source entropy}
 4: $H_{scaled} = \texttt{sigmoid}(H_{diff}/\sqrt{h_{max}})$ {scale between [0,1]}
 5: Init. $T_{vec}$
 6: **for** $h_i \in H_{scaled}$ and $z_i \in Z$ **do**
 7:     $t_i = t_{min} + h_i \cdot (t_{max} - t_{min})$
 8:     $\gamma_i = \frac{\kappa}{\|z_i\|_2}$ {Scale the logit norm}
 9:     $\tau_i = \gamma_i \cdot t_i$ {Adjust regularizer via logit norm}
10:     Store $T_{vec} \leftarrow \tau_i$
11: **end for**
12: **return** $T_{vec}$

---

**Algorithm 2** Dynamic Entropy Control (DEC)

**Input**: $f_{te}, f_s, h_0, X, \kappa$
**Parameters**: $t_{min}, t_{max}, h_{max}, \lambda$
**Output**: $f_s^+$
 1: $T_{vec} = \texttt{CCR}(f_{te}(X), h_0, \kappa, h_{max}, t_{min}, t_{max})$
 2: Init $loss$
 3: **for** $x_i \in X$ and $\tau_i \in T_{vec}$ **do**
 4:     $s_{si} = \texttt{SoftMax}(f_s(x_i))$
 5:     $s_{ti} = \texttt{SoftMax}\left(\frac{f_{te}(x_i)}{\tau_i}\right)$ {smoothen teacher labels}
 6:     $l = (\texttt{avg}(T_{vec}))^2 \cdot \texttt{CE}(s_{si}, s_{ti})$
 7:     Store $loss \leftarrow l$
 8: **end for**
 9: $L = \texttt{avg}(loss)$
10: $f_s' \leftarrow \theta_s - \lambda \nabla L(\theta_s)$
11: $f_s^+ = \texttt{Temperature\_Scale}(f_s', \texttt{avg}(T_{vec}))$
12: **return** $f_s^+$

---

Dynamic Entropy Control calibrates model predictions under distribution shift by adaptively controlling the sharpness of pseudo-labels. Instead of applying a fixed or global temperature, DEC adjusts the certainty of each pseudo-label using a scalar we call the certainty regularizer. This scalar is computed per observation, based on how uncertain the model appears on that input–taking into account both the predicted entropy and the norm of the logit vector. This mechanism allows DEC to smooth unreliable predictions more aggressively, helping the model avoid overconfident errors during test-time adaptation.

## 4.1 The Certainty Regularizer

The key component of DEC is the certainty regularizer, $\tau_i$, a per-sample temperature scalar returned by the `Compute Certainty Regularizer` routine (Algorithm 1). This scalar adjusts how confidently the model treats each pseudo-label. To compute $\tau_i$, we first compare the model's entropy on an input $x_i$ to its typical behavior on the source domain. This reference entropy, $h_0 = \mathbb{E}[\texttt{Entropy}_2(\texttt{SoftMax}(f_{te}(X_s)))]$, reflects the average certainty of the teacher model before adaptation. The difference $H(x_i) - h_0$ is then passed through a sigmoid that is scaled using $h_{max} = \log_2(C)$, the maximum possible entropy over $C$ classes. This yields a

normalized "entropy contrast" in $[0, 1]$, which is mapped to a temperature range defined by $t_{min}$ and $t_{max}$. These bounds control the minimum and maximum smoothing applied to any sample.

To further refine this adjustment, DEC incorporates logit-norm awareness by modulating the entropy-derived temperature with the inverse norm of the model's logits. Specifically, for each sample, we compute the $\ell_2$ norm of its logit vector $\mathbf{z}_i = f_{te}(x_i)$ and scale the temperature accordingly. The scaling factor is given by $\gamma_i = \kappa / |\mathbf{z}_i|_2$, where $\kappa$ is the median logit norm computed over the source-domain samples. This has the effect of increasing $\tau_i$ when the model's confidence is low in logit space, thereby injecting more smoothing into uncertain predictions. The final certainty regularizer is then given by $\tau_i = t_i \cdot \gamma_i$, combining both entropy contrast and logit-norm awareness to calibrate confidence per sample:

$$\tau_i = \underbrace{(t_{\min} + \texttt{sigmoid}\left(\frac{H(x_i) - h_0}{\sqrt{h_{\max}}}\right)(t_{\max} - t_{\min}))}_{\text{entropy contrast } (t_i)} \quad \cdot \quad \underbrace{\left(\frac{\kappa}{\|\mathbf{z}_i\|_2}\right)}_{\text{logit-norm awareness } (\gamma_i)}.$$

Here, $H(x_i)$ is the entropy of the model on $x_i$, $h_0$ is the average entropy on source-domain samples, $\mathbf{z}_i$ is the logit vector, and $\kappa$ is the median logit norm from the source domain.

## 4.2 Addressing Calibration and Mitigating Pseudo-Label Noise

Our formulation allows us to interpret DEC as implicitly minimizing a calibration-aware surrogate. We named $\tau$ the *certainty regularizer* because it regulates the "sharpness" of predicted probabilities and smoothens the pseudo labels produced by the teacher. As $\tau$ increases, the certainty of the prediction decreases. In other words, $\tau$ should be greater for less reliable observations. This $\tau_i$ is used to smooth teacher predictions via temperature-scaled softmax, and the loss becomes:

$$L_{\text{DEC}} = -\sum_{y \in \mathcal{C}} s_i^{\text{teacher}}(y) \log s_i^{\text{student}}(y),$$

where $s_i^{\text{teacher}} = \texttt{Softmax}(f_{te}(x_i)/\tau_i)$ and $s_i^{\text{student}} = \texttt{Softmax}(f_s(x_i))$.

This design has three key calibration properties:

1. **Entropy contrast:** The term $H(x_i) - h_0$ gauges how surprising a prediction is compared to source-domain behavior.

2. **Logit-norm awareness:** Smaller $\|\mathbf{z}_i\|$ values yield higher $\tau_i$, adding smoothing when confidence is low in logit space.

3. **Sample-wise label smoothing:** The softened teacher predictions act as a regularizer against overconfident or noisy pseudo-labels.

Thus, DEC behaves like an adaptive version of label smoothing, where each pseudo-label is calibrated on-the-fly based on its reliability. This implicit regularization helps discourage overconfident mispredictions and explains the empirical gains we observe in ECE and NLL.

To improve calibration after adaptation, we apply a final temperature scaling step: $\texttt{Temperature\_Scale}(f'_s, \texttt{avg}(T_{vec}))$. This temperature is computed from unlabeled test data, preserving the unsupervised setting. It adjusts the student's logits uniformly, smoothing predictions based on the average certainty regularizer. Because $\tau$ guides the temperature scaling component of our algorithm, we follow the example of Guo et al. (2017) and set $t_{min/max}$ within $[1.0, 3.0]$. We recommend setting both these parameters to be higher if the expected domain shift is greater. To show how DEC regularizes observations appropriately, we continue from Example 1 to Example 2:

**Example 2.** *Given the same $g_{t1}, g_{t2}$ and $g_s$ from Example 1, we input these into our* CCR *algorithm. We set $h_0 = \texttt{Entropy}_2(\texttt{SoftMax}(g_s)), \kappa = |g_s|_2, t_{min} = 1.0$ and $t_{max} = 2.0$. Our algorithm first computes a*

*scaled entropy, $H_{scaled}$ with respect to the source domain entropy for $g_s, g_{t1}$ and $g_{t2}$, as 0.49, 0.48, and 0.50, respectively.*

*In step 7 of* CCR, *this entropy is transformed into a preliminary regularizer, $t_i$. Then, step 9 adjusts $t_i$ by considering logit norm with respect to the source-domain logit norm.*

$$\tau_{g_s} = 1.49; \quad \tau_{g_{t2}} = 7.42; \quad \tau_{g_{t1}} = 1.83$$

Notice that, unlike purely entropy-based methods, the CCR algorithm correctly assigns greater regularization to the less reliable samples. Namely, step 9 ensures that samples that are low-entropy due to degenerate reasons are properly regularized by considering logit norm. Furthermore, unique from existing algorithms, our regularizer directly addresses model certainty. The impact of CCR is analyzed in Fig. 3.

An interpretation as to how our model improves accuracy is through the works of Xie et al. (2020) and Lukasik et al. (2020). Although the former's work concerns itself in the semi-supervised learning setting, we found their observations to be relevant. That is, they introduce the *noisy student,* a network that has been *noised* by dropout and stochastic-depth. They find that their noisy student can even learn to outperform the teacher which initially produced the pseudo-labels. For the latter work, they establish that label smoothing mitigates label noise, which is a desirable property with respect to unsupervised adaptation. Specifically, they find that label smoothing can be thought of as a regularizer. This motivates us to smooth more aggressively when we suspect that an observation might be less reliable.

## 5 Experiments

### 5.1 Experimental Setting

In order to evaluate DEC, we conduct a series of experiments using three different backbone models across four datasets. Our primary evaluation metrics will be model accuracy, NLL and $ECE_{bins=15}$ on the observations, allowing us to examine both the predictive performance and the calibration quality of the models. By using varied domains and different backbone architectures, we aim to demonstrate the robustness and adaptability of our algorithm in handling diverse and challenging TTA scenarios. All experiments are run three times. Note that the $t_{min/max}$ hyperparameters are set per dataset; we do not set these per domain shift. We present accuracy and calibration for all datasets along with experiment variances either in the main paper or Appendix A.5 & A.6.

We compare with TENT, T3A, SoTTA and ETA, four recent TTA algorithms which follow the test-time entropy reduction paradigm. We do a single iteration of adaptation for all algorithms unless stated otherwise. Dataset preprocessing steps and a discussion of hyperparameters for all algorithms are in more detail in the Appendix A.2.

### 5.2 Datasets

The following publicly available TTA datasets are used in our experiments; we selected these because they are commonly used in existing works and provide a variety of domain shifts. In total, we evaluate our algorithm over 26 domain shifts which include a total of 282 classes. Furthermore, 15 of our domain shifts have 5 corruption levels. For some datasets, we tested using the "leave one out" (LOO) paradigm; for example, in PACS, to test generalization to *pictures*, we first trained our backbone networks on *art, cartoon, sketch* before adapting.

1. PACS Li et al. (2017) has 4 domains: *pictures, art, cartoon, sketch* with 7 classes. Tested using LOO.

2. HomeOffice Venkateswara et al. (2017) has 4 domains: *art, clipart, product, real* with 65 classes. Tested using LOO.

3. Digits is a combination of 3 "numbers" datasets: USPS Hull (1994), MNIST LeCun et al. (2010), and SVHN Netzer et al. (2011). There are 10 classes. Tested using LOO by training on the source domains' training sets and adapting to target domain's test set.

4. TinyImageNet-C (TIN-C) Le & Yang (2015), has 15 domains with 200 classes. Backbones are trained on corruption-free (source) training set, adapted to and evaluated on corrupted (target) domains. For each target domain, there are 5 tiers of corruption.

### 5.3 Back Bones and Training Details

We test all but the Digits dataset on two popular classifiers, EfficientNetB0 Tan & Le (2019) and MobileNet Howard et al. (2017) pre-trained for ImageNet Deng et al. (2009). We flatten the output of both networks and add a final dense layer with an output shape equivalent to the number of classes.

We evaluate the Digits dataset using "SmallCNN", which is a minor adaptation of the original LeNet LeCun et al. (1998) to include batch normalization layers and max pooling. This serves to represent more compact and straightforward architectures for less complex datasets. The specific details and orderings of the layers in SmallCNN are elaborated on in Appendix A.7. Note that all three models use batch normalization layers as necessitated by ETA, TENT, and SoTTA.

## 6 Results

We present our accuracy, ECE and NLL measurements on the four aforementioned datasets. To show evidence of the over-certainty phenomenon, we also report prediction entropy. More comprehensive figures/tables can be found in the appendix. To show the impact of our `CCR` algorithm, which produces our certainty regularizer $\tau$, we perform an ablation experiment in Fig. 3.

Table 2: Comparison of Shannon entropy, ECE, and NLL averaged across the 15 domain shifts of TIN-C. We use the EfficientNet backbone. Standard deviations across domains are shown as subscripts. This experiment highlights how excessive certainty (low Shannon entropy) correlates with sub-optimal calibration. Our reduction in ECE and NLL is statistically significant ($p < 0.01$) while maintaining competitive accuracy (see Tables 6 and 8).

| Algorithm | Shannon Entropy | | | | | ECE (lower is better) | | | | | NLL (lower is better) | | | | |
|---|---|---|---|---|---|---|---|---|---|---|---|---|---|---|---|
| | Tier 1 | Tier 2 | Tier 3 | Tier 4 | Tier 5 | Tier 1 | Tier 2 | Tier 3 | Tier 4 | Tier 5 | Tier 1 | Tier 2 | Tier 3 | Tier 4 | Tier 5 |
| No Adapt | $2.00_{0.17}$ | $2.10_{0.22}$ | $2.25_{0.30}$ | $2.44_{0.51}$ | $2.63_{0.73}$ | $0.23_{0.02}$ | $0.26_{0.03}$ | $0.29_{0.03}$ | $0.32_{0.04}$ | $0.34_{0.04}$ | $3.06_{0.31}$ | $3.40_{0.45}$ | $3.92_{0.70}$ | $4.57_{0.95}$ | $5.15_{1.07}$ |
| DEC (ours) | $4.48_{0.25}$ | $4.68_{0.34}$ | $4.90_{0.53}$ | $5.01_{0.77}$ | $5.32_{0.80}$ | $\mathbf{0.08}_{0.01}$ | $\mathbf{0.08}_{0.02}$ | $\mathbf{0.07}_{0.02}$ | $\mathbf{0.06}_{0.02}$ | $\mathbf{0.04}_{0.03}$ | $\mathbf{2.27}_{0.12}$ | $\mathbf{2.44}_{0.20}$ | $\mathbf{2.69}_{0.35}$ | $\mathbf{3.04}_{0.59}$ | $\mathbf{3.38}_{0.68}$ |
| T3A | $1.36_{0.11}$ | $1.41_{0.14}$ | $1.46_{0.17}$ | $1.60_{0.20}$ | $1.74_{0.23}$ | $0.33_{0.03}$ | $0.35_{0.03}$ | $0.38_{0.03}$ | $0.41_{0.04}$ | $0.43_{0.04}$ | $3.63_{0.39}$ | $4.09_{0.72}$ | $5.10_{1.80}$ | $7.13_{4.41}$ | $9.82_{7.09}$ |
| ETA | $1.37_{0.08}$ | $1.47_{0.14}$ | $1.57_{0.23}$ | $1.71_{0.38}$ | $1.87_{0.67}$ | $0.37_{0.03}$ | $0.40_{0.03}$ | $0.41_{0.04}$ | $0.43_{0.04}$ | $0.44_{0.04}$ | $4.45_{0.52}$ | $4.93_{0.72}$ | $4.89_{0.64}$ | $5.63_{0.79}$ | $6.29_{1.20}$ |
| TENT | $1.40_{0.08}$ | $1.48_{0.12}$ | $1.57_{0.19}$ | $1.62_{0.33}$ | $1.79_{0.49}$ | $0.27_{0.02}$ | $0.30_{0.03}$ | $0.33_{0.03}$ | $0.36_{0.04}$ | $0.39_{0.04}$ | $3.01_{0.22}$ | $3.36_{0.41}$ | $3.89_{0.75}$ | $4.54_{1.14}$ | $5.20_{1.33}$ |
| SoTTA | $1.71_{0.07}$ | $1.78_{0.12}$ | $1.88_{0.18}$ | $2.08_{0.31}$ | $2.26_{0.41}$ | $0.20_{0.02}$ | $0.21_{0.02}$ | $0.23_{0.03}$ | $0.25_{0.03}$ | $0.28_{0.03}$ | $2.56_{0.13}$ | $2.75_{0.23}$ | $3.04_{0.40}$ | $3.49_{0.75}$ | $3.98_{1.04}$ |

### 6.1 DEC Reduces Calibration Error

Due to our algorithm addressing the over-certainty phenomenon, we significantly improve calibration performance. DEC achieves state-of-the-art average ECE and NLL in all tested datasets and in nearly all individual domain shifts. We recognize that reducing entropy *did* improve calibration compared to baseline in some cases, but the resulting calibration was still sub-optimal. Fig. 3 empirically validates our finding that an adaptive certainty regularizer aids in reducing ECE and NLL. Moreover, the variance of $\tau$ increases as the corruption intensity increases; indicating a broader dynamic range of regularization when encountering more difficult observations.

### 6.2 DEC Augments Accuracy

In addition to strong calibration performance, DEC provides consistent accuracy uplifts while not necessitating any transformations on observations. By jointly exploiting backbone entropy and logit norm (see Fig. 2), we are able to effectively assess observation reliability. After doing so, we apply greater regularization to

less reliable observations. This, in turn, allows us to mitigate the potential label noise produced by the pseudo-labels.

Tables 6, 4 and 8 show that our algorithm maintains competitive accuracy with recent test-time entropy minimization approaches while addressing the over-certainty phenomenon. Moreover, unlike SoTTA and T3A, we do not store observations or training samples during the adaptation process as this could potentially cause security or privacy issues during deployment.

Table 3: Average accuracy, ECE, entropy, and NLL on Digits and PACS datasets tested with LOO. Our approach achieves the best average ECE and NLL. Domain-to-domain $\sigma_{\max}^2$ values for accuracy, ECE, entropy, and NLL are reported.

| Algorithm | Accuracy | ECE | Entropy | NLL |
|---|---|---|---|---|
| No Adapt | 0.589 | 0.301 | 0.439 | 4.465 |
| DEC (ours) | **0.648** | **0.169** | 1.220 | **1.422** |
| T3A | 0.625 | 0.271 | 1.873 | 1.889 |
| ETA | 0.646 | 0.270 | 0.356 | 3.779 |
| TENT | 0.645 | 0.262 | 0.398 | 3.252 |
| SoTTA | 0.640 | 0.252 | 0.474 | 3.222 |
| $\sigma_{\max}^2$ | 0.220 | 0.180 | 0.200 | 46.240 |

Table 4: Performance on the Digits dataset using the SmallCNN backbone.

| Algorithm | Accuracy | ECE | Entropy | NLL |
|---|---|---|---|---|
| No Adapt | 0.873 | 0.105 | 0.081 | 1.182 |
| DEC (ours) | 0.879 | **0.061** | 0.233 | **0.466** |
| T3A | **0.897** | 0.084 | 0.065 | 0.866 |
| ETA | 0.878 | 0.102 | 0.071 | 1.213 |
| TENT | 0.880 | 0.101 | 0.072 | 1.160 |
| SoTTA | 0.887 | 0.096 | 0.062 | 1.188 |
| $\sigma_{\max}^2$ | 0.020 | 0.014 | 0.075 | 1.848 |

Table 5: Performance on the PACS dataset using the EfficientNet backbone.

Table 6: Accuracy on TIN-C with MobileNet backbone across different tiers of corruption. Standard deviations across the domain shifts are shown as subscripts.

| Algorithm | Tier 1 | Tier 2 | Tier 3 | Tier 4 | Tier 5 |
|---|---|---|---|---|---|
| No Adapt | $0.27_{0.04}$ | $0.23_{0.05}$ | $0.19_{0.07}$ | $0.15_{0.08}$ | $0.12_{0.08}$ |
| DEC (ours) | $\mathbf{0.39}_{0.02}$ | $\mathbf{0.37}_{0.03}$ | $\mathbf{0.33}_{0.05}$ | $\mathbf{0.29}_{0.07}$ | $0.24_{0.08}$ |
| T3A | $0.27_{0.04}$ | $0.24_{0.05}$ | $0.20_{0.07}$ | $0.16_{0.08}$ | $0.13_{0.08}$ |
| ETA | $0.22_{0.05}$ | $0.19_{0.06}$ | $0.14_{0.07}$ | $0.11_{0.07}$ | $0.08_{0.06}$ |
| TENT | $0.37_{0.03}$ | $0.34_{0.04}$ | $0.30_{0.06}$ | $0.25_{0.08}$ | $0.20_{0.09}$ |
| SoTTA | $\mathbf{0.39}_{0.02}$ | $\mathbf{0.37}_{0.03}$ | $\mathbf{0.33}_{0.04}$ | $\mathbf{0.29}_{0.06}$ | $\mathbf{0.25}_{0.08}$ |

## 6.3 Discussion

We would like to underscore our algorithm's robustness to the choice of $t_{min/max}$. We performed our experiments by selecting our hyperparameters per dataset, not per domain shift. As shown in Tables 2 and 7, we achieve statistically significant reduction in ECE despite using the same hyperparameters across 15 domain shifts. Our results in Table 1 further bolster our claims of robustness. Again, our hyperparameters are fixed across the four domain shifts but we still achieve statistically significant improvement. In the interest of reproducibility, we release our code at `https://github.com/FinAminToastCrunch/DynamicEntropyControl`.

Our study identifies the *over-certainty phenomenon* of modern TTA methodologies which cause harm to model calibration. To ameliorate this issue, we introduce a certainty regularizer, $\tau$, that modulates model entropy and mitigates pseudo-label noise. The resulting algorithm, DEC, jointly improves model accuracy and reduces calibration error. DEC does not require batch normalization layers like SoTTA, TENT, and ETA do. This permits greater freedom when choosing a backbone. Furthermore, DEC is compatible with existing prototypical learning approaches. We speculate that there is an additional impact of identifying the over-certainty phenomenon: since existing work relies heavily on entropy-based techniques for assessing reliability, we surmise that improving calibration could engender improved reliability estimates. A limitation of our work is that we focus only on the classification setting; we make no claim on whether or not the over-certainty phenomenon occurs in regression scenarios.

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

# A    Appendix

## A.1    Experimental Setup Details

We used TensorFlow 2.9 Abadi et al. (2015) with Nvidia CUDNN version 11.3 on an RTX 3080 16GB laptop GPU with 32GB of system memory. All experiments are run three times using `random_seed` = $0, 1, 2$, respectively.

1. PACS Li et al. (2017) has 4 domains: *pictures, art, cartoon, sketch* with 7 classes. All images are resized to $(227, 227, 3)$ and scaled between $[0, 255]$. Tested using LOO. For both MobileNet and EfficientNet: We set $t_{min}$ and $t_{max}$ parameters to 1.00 and 3.0 respectively.

2. HomeOffice Venkateswara et al. (2017) has 4 domains: *art, clipart, product, real* with 65 classes. All images are resized to $(128, 128, 3)$ and scaled between $[0, 255]$. Tested using LOO. MobileNet: We set $t_{min}$ and $t_{max}$ parameters to 1.20 and 2.75 respectively. EfficientNet: We set $t_{min}$ and $t_{max}$ parameters to 1.00 and 1.75 respectively.

3. Digits is a combination of 3 "numbers" datasets: USPS Hull (1994), MNIST LeCun et al. (2010), and SVHN Netzer et al. (2011). The images are resized to $(32, 32, 1)$ and scaled between $[0, 255]$. There are 10 classes. Tested using LOO by training on the source domains' training sets and adapting to target domain's test set. SmallCNN: We set $t_{min}$ and $t_{max}$ to 1.20 and 2.75, respectively.

4. TinyImageNet-C (TIN-C) Le & Yang (2015), has 15 domains with 200 classes. All images are resized to $(256, 256, 3)$ and scaled between $[0, 255]$. Backbones are trained on corruption-free (source) training set, adapted to and evaluated on corrupted (target) domains. For each target domain, there are 5 tiers of corruption. MobileNet: We set $t_{min}$ and $t_{max}$ parameters to 1.50 and 3.00 respectively. EfficientNet: We set $t_{min}$ and $t_{max}$ parameters to 1.00 and 2.50 respectively.

## A.2    Hyperparameters

We do most initial training on the source domain using `RMS_Prop(lr = 2e − 4)` Tieleman et al. (2012) to minimize cross-entropy loss for `epochs` = $\{15, 15, 5, 25\}$ for each enumerated dataset, respectively. SmallCNN is compiled and initially trained with the Adam optimizer Kingma & Ba (2014) in lieu of RMSProp. We estimate roughly 1,000 hours of GPU usage at 130 watts of power to conduct our experiments. Note that MobileNet expects inputs to be prepossessed in a unique manner. We use Tensorflow's off-the-shelf pre-processing layer for MobileNet at the input. For the tests of statistical significance, we compared DEC with the second best algorithm for each experiment.

**Note that the following are parameters for what we compare against. Further clarification on their meaning can be found in their respective works**. For ETA, we set `E_0 = 0.4 · ln(C)`, as this was their recommended value, and $\epsilon = \{0.6, 0.1, 0.4, 0.125\}$ for each enumerated dataset, respectively. These $\epsilon$ values were empirically chosen to help their performance. For T3A, we set the number of supports to retain, $M = \infty$, as this provides the lowest calibration error. For SoTTA, we set $\rho = 0.05$, $\mathcal{C}_0 = \{0.33, 0.33, 0.33, 0.66\}$ for each dataset respectively to help their performance, and $N_{SoTTA} = 64$ as per their recommendations. We use the authors' recommended batch sizes for all techniques.

We selected hyper parameters either by the recommendation of the respective authors' or in order to improve performance with respect to calibration error. For example, for ETA we experimented with various values for $\epsilon$ in order to lower their ECE as much as possible per dataset (not per domain shift). A similar process was used for selecting $\mathcal{C}_0$ to aid SoTTA.

For our own hyper parameters, we started with $t_{min} = 1.0$ and $t_{max} = 3.0$, then tuned them as we did the other works. We want to emphasize the robustness of our algorithm to the selection of $t_{min/max}$. **Our hyperparameters are chosen for each dataset, not for individual domain shifts.** As shown in Tables 2 and 7, we achieve a statistically significant reduction in ECE and NLL despite using consistent

hyperparameters across **15 domain shifts**. Our results in Table 1 further support our claims of robustness, showing a statistically significant improvement even with fixed hyperparameters across the four domain shifts. Another note is that although step 6 of DEC (Algorithm 2) shows the teacher model inferencing, we can actually just store and reuse the logits computed in line 1 of the `CCR` Algorithm 1. This saves us one forward pass computation. We use a batch size of 50 for our DEC for all experiments.

### A.3 Further Discussion of Related Work

As stated previously, the focus of our work is on TTA algorithms. However, we would like to discuss some studies on OOD calibration to provide supplementary background. The authors of Wang et al. (2020b) and Gong et al. (2021) remark on poor calibration in the OOD setting, however, according to their algorithms, they require the unlabeled-domain-shifted observations **during training**. That is, their algorithm is not for the deployed setting like ours is. Another work, Wald et al. (2021), discusses proxies for determining if a model would be calibrated during deployment, before deployment. Finally, there exist works such as Hu et al. (2024) which are post-hoc calibration techniques (not TTA algorithms). We recognize the relevance of these works but emphasize that they address different facets or settings.

Recent work by Press et al. (2024) offers a complementary view on the failure modes of entropy minimization under domain shift. Their study shows that while early adaptation clusters test embeddings near training data and boosts accuracy, continued optimization pushes embeddings away—leading to accuracy collapse. This biphasic behavior helps explain why TTA methods often degrade in calibration over time. While their focus is on representation drift and unsupervised accuracy estimation, our work highlights a parallel failure mode: over-adaptation also induces epistemic miscalibration.

### A.4 Memory Tradeoffs

The memory overhead of T3A, $P_{T3A}$, is a function of their hyperparameter $M$ and the size of the final classifier. For SoTTA, the memory overhead is the same as TENT/ETA plus the $N_{SoTTA}$ observations it stores. More formally, by defining $\psi_\omega, \phi_\omega$ as the number of parameters in the feature extractor and classifier, respectively:

$$P_{T3A} = M \cdot \phi_\omega \tag{7}$$

$$P_{DEC} = 2 \cdot (\psi_\omega + \phi_\omega) \tag{8}$$

$$P_{ETA} = P_{TENT} = \texttt{NumBNParams} \tag{9}$$

$$P_{SoTTA} = \texttt{NumBNParams} + N_{SoTTA} \cdot |x| \tag{10}$$

Note that the frozen copy of the backbone (teacher model) used in DEC does not need to be in GPU VRAM at the same time as the backbone being adapted.

## A.5 Additional Experiments

Table 7: Comparison of Shannon entropy, ECE, and NLL averaged across the 15 domain shifts of TIN-C. We use the MobileNet backbone. Standard deviations across domains are shown as subscripts. This experiment highlights how excessive certainty (low Shannon entropy) correlates with sub-optimal calibration. Our reduction in ECE and NLL is statistically significant ($p < 0.01$) while maintaining competitive accuracy.

| Algorithm | Shannon Entropy | | | | | ECE (lower is better) | | | | | NLL (lower is better) | | | | |
|---|---|---|---|---|---|---|---|---|---|---|---|---|---|---|---|
| | Tier 1 | Tier 2 | Tier 3 | Tier 4 | Tier 5 | Tier 1 | Tier 2 | Tier 3 | Tier 4 | Tier 5 | Tier 1 | Tier 2 | Tier 3 | Tier 4 | Tier 5 |
| No Adapt | $1.44_{0.10}$ | $1.53_{0.14}$ | $1.61_{0.18}$ | $1.67_{0.19}$ | $1.66_{0.21}$ | $0.43_{0.02}$ | $0.44_{0.03}$ | $0.47_{0.05}$ | $0.50_{0.06}$ | $0.53_{0.08}$ | $5.98_{0.55}$ | $6.56_{0.88}$ | $7.36_{1.41}$ | $8.42_{2.04}$ | $9.50_{2.66}$ |
| DEC (ours) | $3.35_{0.21}$ | $3.55_{0.30}$ | $3.80_{0.42}$ | $3.97_{0.42}$ | $4.04_{0.38}$ | $\mathbf{0.05}_{0.01}$ | $\mathbf{0.05}_{0.02}$ | $\mathbf{0.06}_{0.02}$ | $\mathbf{0.07}_{0.03}$ | $\mathbf{0.10}_{0.06}$ | $\mathbf{2.68}_{0.11}$ | $\mathbf{2.83}_{0.17}$ | $\mathbf{3.05}_{0.29}$ | $\mathbf{3.37}_{0.46}$ | $\mathbf{3.78}_{0.78}$ |
| T3A | $1.60_{0.10}$ | $1.70_{0.14}$ | $1.81_{0.17}$ | $1.87_{0.17}$ | $1.89_{0.21}$ | $0.42_{0.02}$ | $0.44_{0.02}$ | $0.46_{0.03}$ | $0.48_{0.05}$ | $0.50_{0.06}$ | $6.36_{0.66}$ | $7.15_{1.10}$ | $9.17_{3.37}$ | $13.27_{7.59}$ | $19.55_{15.24}$ |
| ETA | $1.19_{0.10}$ | $1.22_{0.10}$ | $1.36_{0.15}$ | $1.32_{0.19}$ | $1.34_{0.16}$ | $0.51_{0.04}$ | $0.54_{0.05}$ | $0.56_{0.05}$ | $0.59_{0.06}$ | $0.62_{0.06}$ | $8.69_{0.98}$ | $8.93_{0.86}$ | $9.83_{2.17}$ | $10.68_{2.02}$ | $12.06_{2.42}$ |
| TENT | $1.10_{0.04}$ | $1.14_{0.06}$ | $1.21_{0.09}$ | $1.28_{0.12}$ | $1.34_{0.12}$ | $0.38_{0.02}$ | $0.40_{0.03}$ | $0.43_{0.04}$ | $0.47_{0.06}$ | $0.50_{0.07}$ | $4.95_{0.35}$ | $5.47_{0.63}$ | $6.25_{1.17}$ | $7.35_{1.94}$ | $8.54_{2.83}$ |
| SoTTA | $1.14_{0.04}$ | $1.19_{0.05}$ | $1.26_{0.09}$ | $1.34_{0.12}$ | $1.43_{0.16}$ | $0.36_{0.01}$ | $0.37_{0.02}$ | $0.39_{0.03}$ | $0.42_{0.04}$ | $0.45_{0.05}$ | $4.70_{0.23}$ | $4.99_{0.37}$ | $5.44_{0.66}$ | $6.10_{1.01}$ | $6.89_{1.53}$ |

Table 8: Accuracy on TIN-C with EfficientNet backbone across different tiers of corruption for various algorithms. Standard deviations are shown as subscripts.

| Algorithm | Tier 1 | Tier 2 | Tier 3 | Tier 4 | Tier 5 |
|---|---|---|---|---|---|
| No Adapt | $0.41_{0.04}$ | $0.37_{0.06}$ | $0.31_{0.08}$ | $0.24_{0.09}$ | $0.19_{0.10}$ |
| DEC (ours) | $\mathbf{0.49}_{0.02}$ | $\mathbf{0.46}_{0.04}$ | $0.41_{0.06}$ | $0.35_{0.10}$ | $0.29_{0.11}$ |
| T3A | $0.42_{0.04}$ | $0.38_{0.06}$ | $0.32_{0.08}$ | $0.26_{0.10}$ | $0.21_{0.10}$ |
| ETA | $0.33_{0.03}$ | $0.29_{0.06}$ | $0.26_{0.07}$ | $0.22_{0.08}$ | $0.19_{0.09}$ |
| TENT | $0.47_{0.03}$ | $0.42_{0.05}$ | $0.37_{0.08}$ | $0.31_{0.10}$ | $0.25_{0.11}$ |
| SoTTA | $\mathbf{0.49}_{0.02}$ | $\mathbf{0.46}_{0.03}$ | $\mathbf{0.42}_{0.06}$ | $\mathbf{0.36}_{0.09}$ | $\mathbf{0.31}_{0.10}$ |

Table 9: Average accuracy, ECE, entropy, and NLL on the Home Office dataset using the EfficientNet backbone. Domain-to-domain $\sigma^2_{\max}$ values for accuracy, ECE, entropy, and NLL are reported.

| Algorithm | Accuracy | ECE | Entropy | NLL |
|---|---|---|---|---|
| No Adapt | 0.665 | 0.198 | 0.642 | 2.184 |
| DEC | 0.669 | **0.069** | 1.832 | **1.490** |
| T3A | **0.686** | 0.267 | 0.186 | 4.321 |
| ETA | 0.665 | 0.218 | 0.422 | 2.745 |
| TENT | 0.684 | 0.211 | 0.460 | 2.396 |
| SoTTA | 0.667 | 0.194 | 0.648 | 2.097 |
| $\sigma^2_{\max}$ | 0.060 | 0.030 | 1.110 | 3.760 |

Table 10: Average accuracy, ECE, entropy, and NLL on the Home Office dataset using the MobileNet backbone. Domain-to-domain $\sigma_{\max}^2$ values for accuracy, ECE, entropy, and NLL are reported.

| Algorithm | Accuracy | ECE | Entropy | NLL |
|---|---|---|---|---|
| No Adapt | 0.567 | 0.242 | 0.824 | 2.184 |
| DEC | 0.575 | **0.046** | 2.183 | **1.686** |
| T3A | **0.617** | 0.335 | 0.182 | 4.321 |
| ETA | 0.553 | 0.254 | 0.807 | 2.745 |
| TENT | 0.579 | 0.242 | 0.776 | 2.441 |
| SoTTA | 0.574 | 0.233 | 0.833 | 2.409 |
| $\sigma_{\max}^2$ | 0.013 | 0.011 | 0.116 | 6.467 |

Table 11: Average accuracy, ECE, entropy, and NLL on the PACS dataset using the MobileNet backbone. Domain-to-domain $\sigma_{\max}^2$ values for accuracy, ECE, entropy, and NLL are reported.

| Algorithm | Accuracy | ECE | Entropy | NLL |
|---|---|---|---|---|
| No Adapt | 0.841 | 0.111 | 0.177 | 1.182 |
| DEC (ours) | 0.848 | **0.082** | 0.341 | **0.466** |
| T3A | **0.857** | 0.116 | 0.096 | 0.866 |
| ETA | 0.842 | 0.101 | 0.192 | 1.213 |
| TENT | 0.848 | 0.108 | 0.162 | 1.160 |
| SoTTA | 0.852 | 0.102 | 0.180 | 1.188 |
| $\sigma_{\max}^2$ | 0.034 | 0.015 | 0.099 | 1.016 |

## A.6 Run-to-Run Variances across all TTA algorithms

- For the Digits dataset: $\sigma_{\max}^2 = [4.39 \times 10^{-3}, 5.51 \times 10^{-3}, 3.85 \times 10^{-3}, 1.00 \times 10^{-3}]$ for accuracy, ECE, and entropy, respectively across all trials using SmallCNN.

- For Home Office across both backbones: $\sigma_{\max}^2 = [5.00 \times 10^{-2}, 1.12 \times 10^{-5}, 1.66 \times 10^{-5}, 7.29 \times 10^{-3}]$ for accuracy, ECE, and entropy, respectively.

- For the PACS dataset across both backbones: $\sigma_{\max}^2 = [1.21 \times 10^{-3}, 1.02 \times 10^{-4}, 2.09 \times 10^{-3}, 2.00 \times 10^{-4}]$ for accuracy, ECE, and entropy, respectively across all trials.

- For the TIN-C dataset across both backbones and across all 5 domain shifts: $\sigma_{\max}^2 = [1.39 \times 10^{-3}, 1.12 \times 10^{-7}, 2.46 \times 10^{-3}, 2.40 \times 10^{-1}]$ for accuracy, ECE, and entropy, respectively across all trials.

## A.7 SmallCNN

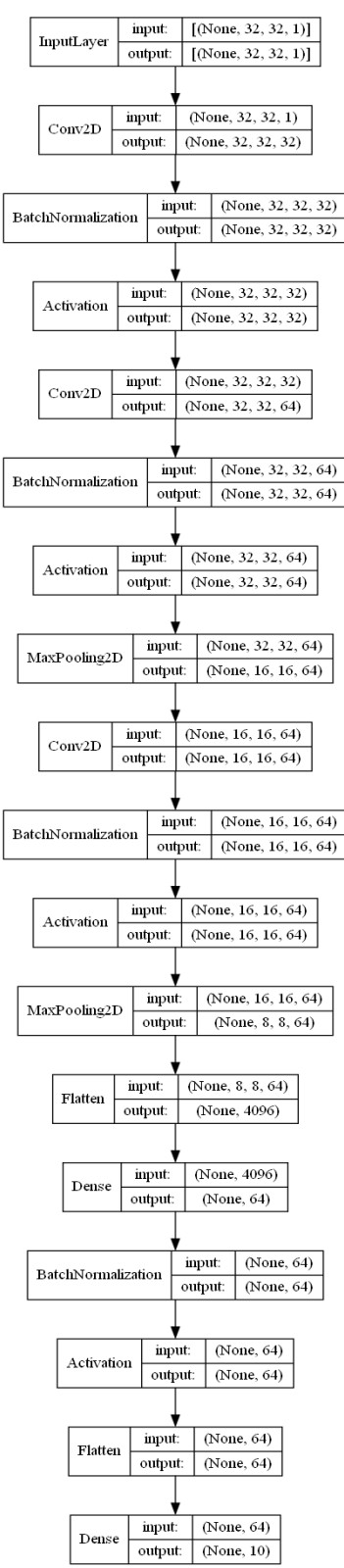

Figure 4: SmallCNN's architecture has 319,498 parameters.

