# OpenReview forum: "The Over-Certainty Phenomenon in Modern Test-Time Adaptation Algorithms"
_TMLR — Accepted by TMLR_

### Review · Reviewer_56nD · 2025-04-28

**Summary Of Contributions:**

The paper identifies and studies the over-certainty phenomenon (OCP) in modern test-time adaptation (TTA) algorithms, where entropy-minimization-based methods unintentionally worsen model calibration under domain shift. While previous work focuses primarily on accuracy gains during TTA, this study is the first to systematically highlight how these methods can drive models to become overly confident, exacerbating calibration errors. To address OCP, the authors propose Dynamic Entropy Control (DEC) — a novel TTA method that dynamically adjusts the confidence of pseudo-labels during adaptation based on the relative entropy and logit norm of observations compared to the source domain. DEC introduces a certainty regularizer that adapts temperature scaling individually per sample. Comprehensive experiments across 26 domain shifts and several datasets (e.g., PACS, HomeOffice, Digits, TinyImageNet-C) show that DEC consistently reduces expected calibration error (ECE) and negative log-likelihood (NLL) while maintaining or improving accuracy. The method is also shown to be robust across datasets with fixed hyperparameters, without the need to store test-time data.

**Audience:**

Yes

**Broader Impact Concerns:**

Impactful Practical Implications: Highlights why calibrated uncertainty is critical in real-world safety-critical applications.

**Claims And Evidence:**

Yes

**Requested Changes:**

1. Add theoretical justification or more intuition for the τ scaling
2. Improve the abstract as mentioned above
3. Report runtime and memory overhead compared to TENT, SoTTA.

**Strengths And Weaknesses:**

## Strengths:
1. Thorough Experiments: Extensive empirical validation across diverse datasets and backbones; statistical significance is reported.
2. Robustness: Demonstrates that DEC performs consistently across different corruption intensities and domains without per-domain tuning.


## Weaknesses:
1. Over-cartainty problem, is often called overconfidence problem has been studied in this field.
2. Lack of novelty, the identification of over-certainty problem is discussed by some works, for example [1]
3. Writing needs to be improved, for example, the abstract should provide enough information for user to understand the work. However, how the method works is not mentioned in the abstract
4. Most analysis are based on heuristic,
5. Is there any instruction for the hyper parameter t_max, t_min?
6. What if the BN is replaced by other things such as layer norm or group norm.





[1] Pseudo-Calibration: Improving Predictive Uncertainty Estimation in Unsupervised Domain Adaptation

---

> ### Author Response · Authors · 2025-05-22
> **Response Part 1: Novelty, Clarity, and Implementation Feedback**
>
> We thank the reviewer for the helpful feedback. Due to the character limit, we respond to the reviewer’s request for additional theoretical intuition in a separate comment.
>
> ### Regarding Novelty and Related Work:
>
> > “Over-c[e]rtainty problem, is often called overconfidence problem has been studied in this field.
> > Lack of novelty, the identification of over-certainty problem is discussed by some works, for example [1]”
>
> We appreciate the reviewer’s reference to prior work. While miscalibration has been explored in domain adaptation (e.g., [1]), our contribution is not in simply observing miscalibration, but in systematically identifying and isolating a failure mode unique to entropy-minimization-based TTA algorithms. Moreover, our paper gives mechanistic insight as to why this happens. To the best of our knowledge, our work is the first to identify that some existing TTA algorithms engender calibration that is worse than if the model had been left alone. To summarize, our contribution is **not** “shifting to a new domain harms backbone model calibration” which **is** an established problem with a plethora of existing solutions—which we discussed in Appendix A.3.
>
> For [1], their focus is on a post-hoc calibration step which happens after adaptation—it is not a standalone TTA algorithm. Their work is motivated by the well-studied tendency of backbone models becoming less calibrated under distribution shift. This is not the focus of our work. Instead, our focus is on a popular design trend observed in TTA literature. With that being said, we will include a discussion of this work in our related works section.
>
> ---
>
> ### Regarding Improvements to Writing:
>
> > "Writing needs to be improved… The abstract should provide enough information... how the method works is not mentioned in the abstract"
>
> We greatly appreciate the reviewer’s feedback on the writing quality. We commit to revising the abstract to include greater clarity on the mechanism of our algorithm. Are there any other components you would like us to rewrite or clarify? We will upload an updated version of our manuscript once all the reviews have been posted.
>
> ---
>
> ### Practicality and Implementation Questions:
>
> > "Is there any instruction for the hyper parameter t_max, t_min?"
>
> For these parameters, we follow the convention of [3] and recommend setting these values between 1.0 and 3.0 respectively. We state this at the bottom of page 7. We recommend setting $t_{\text{min}}$ and $t_{\text{max}}$ to be higher if the expected domain shift is greater. We will update our manuscript to make the instructions for these hyperparameters more clear.
>
> > "What if the BN is replaced by other things such as layer norm or group norm?"
>
> Thank you for raising this point. Indeed, this was a question we considered during experimentation. However, a key challenge is that the primary methods we compare against—including TENT, ETA, and SoTTA—are designed specifically around updating batch normalization (BN) statistics during test-time adaptation. Their performance relies on the presence of BN layers, and altering them to use LayerNorm or GroupNorm would invalidate their original design and likely compromise the baselines.
> Thus, to ensure a fair comparison, we retained the BN configuration used in the original implementations. Evaluating alternative normalization strategies is an interesting direction, but would require reworking each baseline to ensure compatibility and fair intent-preserving comparisons—which is not the scope of this paper and we leave for future work.
>
> > "Report runtime and memory overhead compared to TENT, SoTTA."
>
> Thank you for this suggestion. We will post an updated version of our manuscript with the memory overhead once all the reviews have been posted. We acknowledge that runtime overhead is an important practical consideration, but direct comparison is nontrivial in our setting: methods like TENT, SoTTA, T3A, and ETA each rely on specific adaptation mechanisms (e.g., updating class templates vs. sharpness-aware minimization) which can have wildly different timing results depending on implementation. There was not a way for us to measure run-time performance that was independent of confounding variables such as the movement of memory between CPU and GPU or compiler optimization.
>
> ---
>
> **References**
> [1] *Pseudo-Calibration: Improving Predictive Uncertainty Estimation in Unsupervised Domain Adaptation*
> [2] *Towards Stable Test-Time Adaptation in Dynamic Wild World*
> [3] Chuan Guo, Geoff Pleiss, Yu Sun, and Kilian Q. Weinberger. *On Calibration of Modern Neural Networks*. In *International Conference on Machine Learning*, pp. 1321–1330. PMLR, 2017.

---

> > ### Author Response · Authors · 2025-05-26
> > **Response Part 2: More intuition for $\tau$ scaling**
> >
> > We thank the reviewer for requesting theoretical intuition. DEC can be viewed as implicitly minimizing a calibration-aware surrogate.
> >
> > First, recall the classical entropy minimization objective used in TENT:
> >
> > $\mathcal{L_{Tent}} = -\sum_{y \in \mathcal{C}} f(y|x) \log f(y|x)$
> >
> > which lacks any regularization and leads to overconfident predictions under distribution shift.
> >
> > In contrast, DEC constructs per-sample temperature scalars $\tau_i$ using both entropy shift and logit norm:
> >
> > We define two components:
> >
> > $entropy \\: adjustment_i$ $ = \left( t_{\min} + \sigma\left( \frac{H(x_i) - h_0}{\sqrt{h_{\max}}} \right)(t_{\max} - t_{\min}) \right)$
> >
> > $logit \\: scaling_i = \left( \frac{\kappa}{\| \mathbf{z}_i \|_2} \right)$
> >
> > $\tau_i = (entropy \\: adjustment_i) \cdot (logit \\: scaling_i)$
> >
> > where:
> >
> > - $H(x_i)$ is the entropy of the model prediction on $x_i$,
> > - $h_0$ is the average entropy on source-domain samples,
> > - $\mathbf{z}_i = f(x_i)$ is the logit vector before softmax,
> > - $\kappa$ is the median $\ell_2$ norm of source-domain logits.
> >
> > This $\tau_i$ is then used to smooth pseudo-labels via temperature-scaled softmax:
> >
> > $s_{teacher \\: i} = Softmax(f_{teacher}(\frac{x_i}{\tau_i})) \quad \text{(a frozen copy of the backbone before adaptation)}$
> >
> > And we compute the predictions of the backbone model:
> >
> > $s_{student\\: i} = Softmax(f_{student}({x_i})) \quad \text{(the backbone that is being adapted)}$
> >
> > and the optimization minimizes our regularized cross-entropy: $$ = -\sum_{y \in \mathcal{C}} s^{teacher}_i(y) \log s^{student}_i(y)$$
> >
> > This has key calibration properties:
> >
> > 1. **Entropy contrast**: If $H(x_i)$ vs. $h_0$, i.e., the model's typical certainty is compared with the certainty of the prediction. Existing approaches rely on entropy without any reference. For example, the authors of ETA reject observations with an entropy greater than $0.4\cdot ln(C)$. This rejection criteria is not informed by source domain certainty.
> >
> > 2. **Logit norm gating**: If $\|\mathbf{z}_i\|$ is small (i.e., low confidence in logit space), $\tau_i$ is increased, which adds smoothing. As we showed in Figure 2 and Example 1, relying purely on entropy to assess reliability of an observation can produce a misleading conclusion.
> >
> > 3. **Sample-wise Label Smoothing**: Our regularized cross-entropy objective softens the teacher predictions to account for lack of reliability on the pseudo-label. This idea is justified by two works: *Does label smoothing mitigate label noise?* and *Self-training with noisy student improves imagenet classification.* These two works explain that softening labels can work as a form of regularization against poor labels.
> >
> > Thus, DEC behaves like a soft, per-sample label smoothing method, analogous to temperature scaling (Guo et al., 2017) but adaptive and online. Rather than applying a global post-hoc calibration, DEC adjusts each pseudo-label in real-time, based on both **relative** entropy and logit norm. This implicitly discourages overconfident mispredictions and improves calibration—matching our observed improvements in ECE and NLL across diverse domain shifts. We will revise the paper to include greater intuition behind the $\tau$ scaling as you suggested. We will send a message once this is done.
> >
> > [4] Xie, Qizhe, et al. "Self-training with noisy student improves imagenet classification." Proceedings of the IEEE/CVF conference on computer vision and pattern recognition. 2020.
> >
> > [5] Lukasik, Michal, et al. "Does label smoothing mitigate label noise?." International Conference on Machine Learning. PMLR, 2020.

---

> > > ### Author Response · Authors · 2025-05-27
> > > **Clarification on \sigma**
> > >
> > > The $\sigma$ in the prior response corresponds with a sigmoid to scale values between 0 and 1.

---

### Review · Reviewer_F8Wi · 2025-05-05

**Summary Of Contributions:**

This paper considers the calibration probelm in the OOD task, which can amplify the over-confidence/under-confidence probelms. Usually, the entropy is used to assess the uncertainty of pseudo-labeling. However, this can have limitations in the lack of comparison with source entropy and a more detailed strategy. The paper proposed a new algorithm addressing the over-confidence/under-confidence probelm in test-time adaptation algorithms. The results look promising, and DEC's algorithm can resolve the over-confidence/under-confidence problem in many datasets and tasks.

**Audience:**

Yes

**Broader Impact Concerns:**

There is no critical issue on Broader Impact.

**Claims And Evidence:**

Yes

**Requested Changes:**

Page 7

- $SoftMax_{T=1}$ and $SoftMax_{T=\tau_i}$ are not defined theruoghly.
- $Temperature-scaling$ $(f^{'}_s,  {avg} (T_vec) )$ is not defined theruoghly.

These two terminologies are typical in the classification task. However, more clarified definitions are required for a more understandable paper.

In Table 6: Accuracy on TIN-C, the accuracy is overall lower; maybe the task can be difficult. In this case, the advantages of DEC in calibration are not well-presented. It is better to present some clues about these advantages.

**Strengths And Weaknesses:**

Strengths
1. This paper studies the calibration probelm in OOD setups, which is critical in real-world scenarios. In addition, the results show an impressive improvement.
2. There are clear motivations for this probelm, and the solution is also valid.

Weaknesses
1. Though the test-time adaptation is the main topic, the conventional calibration approaches, such as temperature scaling and in-processing algorithms, can be baselines that are not thoroughly studied.
2. In the algorithms, there are some terms not clarified, which raises difficulties in understanding the results.

---

> ### Author Response · Authors · 2025-05-16
> **Clarifications on Softmax Temperature Notation, Calibration Baselines, and TIN-C Results**
>
> > *These two terminologies are typical in the classification task. However, more clarified definitions are required for a more understandable paper… SoftMaxT=1 and SoftMaxT=τᵢ are not defined [thoroughly]… Temperature−scaling (fs′,avg(Tvec)) is not defined [thoroughly].*
>
> Thank you for this helpful comment. We agree that additional clarification would improve the readability of this section. In the revised manuscript, we will make the following adjustments:
>
> SoftMaxT=1 and SoftMaxT=τᵢ refer to the softmax function applied with temperature `T`. Specifically, given logits `z`, the temperature-scaled softmax is defined as:
>
>     SoftMax_T(z_i) = exp(z_i / T) / sum_j exp(z_j / T)
>
> In our notation, `SoftMaxT=1` refers to the standard softmax, and `SoftMaxT=τᵢ` refers to using a per-sample adaptive temperature `τᵢ` computed by the CCR algorithm.
>
> Temperature-scaling `(f_s′, avg(Tvec))` refers to the final smoothing step at the end of Algorithm 2, where we apply temperature scaling using the mean `τ` value across the batch to calibrate the adapted model `f_s′`. This mirrors the idea of temperature scaling from Guo et al. (2017), adapted here for test-time use without target labels.
>
> We will make these definitions explicit in the main text and ensure all symbols are clearly introduced at first use to improve the paper’s clarity. We will send a message once we update the manuscript to reflect this change.
>
> > *Though the test-time adaptation is the main topic, the conventional calibration approaches, such as temperature scaling and in-processing algorithms, can be baselines that are not thoroughly studied.*
>
> We would like to emphasize that conventional calibration techniques such as post-hoc temperature scaling or label smoothing require access to labeled target data, which is not available in the TTA setting. There do exist unsupervised calibration techniques which we have already discussed in Appendix (A.3).
>
> As our goal is to develop a test-time adaptation method that incorporates calibration as a built-in objective—rather than as a post-processing step—we did not position our work to cover standalone calibration techniques. That said, we do provide a direct comparison between our certainty regularizer (CCR) and naive temperature scaling in Figure 3.
>
> > *In Table 6: Accuracy on TIN-C, the accuracy is overall lower; maybe the task can be difficult. In this case, the advantages of DEC in calibration are not well-presented. It is better to present some clues about these advantages.*
>
> The accuracy engendered by our algorithm is never lower than No Adapt. Perhaps there was a misunderstanding, Table 6 shows that DEC performs comparably to SoTTA, the existing state of the art, in all but the highest tier of corruption (tier 5). In tier 5, our accuracy is 0.24 compared to the 0.25 of SoTTA.

---

### Review · Reviewer_iKPA · 2025-05-18

**Summary Of Contributions:**

The paper identifies the "over-certainty phenomenon" in test-time adaptation algorithms, where entropy minimization leads to miscalibrated predictions under domain shift. It proposes Dynamic Entropy Control (DEC), which improves model calibration by adaptively adjusting prediction certainty based on entropy and logit norm analysis. Extensive experiments across domain shifts demonstrate state-of-the-art calibration performance (measured by ECE and NLL) while maintaining competitive accuracy. The work provides both mechanistic insight into why current TTA methods struggle with calibration and a practical solution to address this limitation.

**Audience:**

Yes

**Claims And Evidence:**

Yes

**Requested Changes:**

Please refer to the Weakness.

**Strengths And Weaknesses:**

**Strength**

- Investigates the calibration issue in entropy-minimizing test-time adaptation algorithms

- Offers mechanistic analysis of model certainty behavior during adaptation

- Provides comprehensive empirical evaluation across multiple datasets, domain shifts, and model architectures

**Weakness**

My main concern lies in two points:

- The experimental setting should be validated on larger dataset benchmarks, as the current evaluation is relatively small. TTA is a lightweight task, and we tend to adapt only very small numbers of trainable parameters, as well as run on each example just one or a few times.

- Authors should explore additional applications of their method in TTA settings where calibration offers greater benefits to highlight their method's importance - for example, applying their approach to improve sample selection in TTA, which currently relies on entropy-based reliability measures

---

> ### Author Response · Authors · 2025-05-22
> **Addressing Concerns on Benchmark Size and Downstream Use Cases**
>
> >“The experimental setting should be validated on larger dataset benchmarks, as the current evaluation is relatively small. TTA is a lightweight task, and we tend to adapt only very small numbers of trainable parameters, as well as run on each example just one or a few times.”
>
> We performed the largest experiments possible with respect to our compute limitations. While it is true that there exist larger benchmarks, we would like to underscore that Tiny-ImageNet-C (TIN-C), one of the benchmarks we used, is not “tiny.” And in fact, it is one of the largest TTA benchmarks used in existing literature. TIN-C contains 750,000 images across 200 classes for evaluation. Additionally, HomeOffice consists of almost 16,000 images across 65 classes. Finally, due to our inclusion of PACS (~10,000 images) and Digits (\~180,000 images), our overall evaluation encompasses over 950,000 images.
>
> We also evaluated three distinct backbone networks, whereas prior work typically focuses on a single architecture such as ResNet26/50. Due to compute limitations and a desire to support reproducibility, we intentionally focused on datasets that strike a balance between scale and accessibility. We believe this design choice strengthens the impact and reach of our results without compromising rigor.
>
>
> >“ Authors should explore additional applications of their method in TTA settings where calibration offers greater benefits to highlight their method's importance — for example, applying their approach to improve sample selection in TTA, which currently relies on entropy-based reliability measures. "
>
>
> We thank the reviewer for this ingenious observation. Indeed, if we have a more calibrated backbone, then the downstream sample selection in TTA would likely perform better! However we believe that the impact of improved calibration on entropy-based reliability measures should be the focus of future work. Our paper’s stated scope is deliberately compact. Adjusting the calibration of prior art to investigate sample-selection performance would easily double the length of the manuscript and blur the take-home message. We therefore treat integration with prior art’s sample-selection mechanisms as valuable follow-up work rather than a core contribution. With that being said, we will add a discussion on the impact of the over-certainty phenomenon on entropy-based reliability measures in the revision.
>
> Regarding additional applications: we gave an overview of potential applications of our work in section 1.1. In this section we overview how our study highlights potential danger cases and biases caused by the over-certainty phenomenon. In summary, we underscore our method’s importance by explaining how overconfident models may disadvantage certain groups or cause unsafety when high-confidence predictions guide resource allocation or risk assessments.

---

> > ### Comment · Reviewer_iKPA · 2025-05-25
> >
> > Thank the authors for their response. Here reviewer wants to clarify what the reviewer aims to further understand:
> >
> > Relating to the experiment: Reviewers request further experiments because reviewers think that this over-confidence comes from some perspective: distribution shift type (as authors have verified on different datasets), and model size (traditional calibration shows that larger model size leads to more over-confidence), if authors have do it, you can further clarify in the experiment section.
> >
> > About the point reviewers want authors to give additional information to highlight the reliability of the current method: Authors indicate that they leave this question for future research, but it is a key point to highlight the proposed method's contribution.  We are working on TTA calibration, a different setting from traditional calibration, so we can not naturally utilize the existing evidence in traditional calibration techniques, which reduces over-confidence, will help model behave more reliably or robustness (except that the author can build a theoretical connection in this setting and the original calibration setting).

---

### Author Response · Authors · 2025-05-28
**Revision Uploaded**

Dear Reviewers,

Thank you for your helpful feedback, we have uploaded a revision of our paper to address the reviewer comments.


For reviewer Reviewer F8Wi:
- We have clarified the softmax and temperature scaling syntax
- We have increased our discussion of conventional and unsupervised calibration techniques in Appendix A.3. We added your suggested citation.
- We explained why TTA is difficult in page 2.

For reviewer 56nD:
- We have updated the abstract to give a more mechanistic explanation as to how our technique works.
- We have added an explanation of t_min and t_max (Section 4)
- We have overhauled section 4 to give greater intuition and mechanistic understanding of our algorithm.
- We added a memory overhead section the appendix (A.4)

For reviewer iKPA:
- In 6.3 we added a discussion which speculates on the improvement in reliability estimates due to improved calibration.

---

### Decision · Action_Editor_EmJr · 2025-06-17

**Recommendation:** Accept as is

**Additional Comments:**

This paper is on the topic of test-time domain adaptation (TTA) and in particular makes the important observation of a "over-certainty" phenomenon inherent in entropy-minimizing TTA algorithms (which are a popular family in TTA), that lead to over certainty in predictions. The author address this by proposing a method that dynamically regulates the certainty of pseudo-labels using a temperature that is determined both by the entropy and the logit norm. The authors provide extensive experiments across multiple datasets and architectures, demonstrating that DEC improves calibration metrics (ECE, NLL) while maintaining competitive accuracy with state-of-the-art TTA methods.

The reviewers found that the problem studied here is critical in real-world scenarios and well-motivated (Reviewer F8Wi), the paper offers a mechanistic analysis of model certainty behavior during adaptation and proposes a method that is effective, as demonstrated through comprehensive empirical evaluation across multiple datasets, domains shifts, and model architectures, recognized by all reviewers.

During the rebuttal, the authors made some clarifications of notation and definitions and the tasks studied, and corrected some misunderstandings. For example, reviewer F8Wi claimed that some important baselines are missing, but the authors clarified that those aren't applicable for the setting of TTA due to requiring labeled data which aren't available in TTA. The authors added further discussion of baselines (in particular unsupervised calibration techniques) to the updated manuscript. The authors have also made improvements to the writing quality, based on feedback from the reviewers and added theoretical intuition to better explain the success of their method (it can be seen as soft per-example smoothing that happens online in an adaptive manner, unlike classic temperature scaling methods).
The authors did not address a suggestion by reviewer iKPA to evaluate whether improved calibration also improves sample selection since they argued that exploring such use cases comprehensively is greatly out of scope of the paper, which seems fair. However, the authors added discussion about this in their revision which can be helpful to guide future work in this direction.

While all reviewers pointed out that the phenomenon of miscalibration has been studied before in machine learning, I agree with the authors that observing this in the context of TTA, identifying it as a failure mode of a particular family of TTA methods and giving mechanistic insight for why it happens in these methods is interesting and practically valuable. Overall, I recommend acceptance as the paper is relevant and interesting to TMLR and the claims are backed by substantial evidence.

**Audience:**

Yes

**Audience Explanation:**

This paper is on the interesting and relevant topic of TTA, in particular on observing an important failure mode of a family of algorithms and proposing an effective solution to address it.

**Claims And Evidence:**

Yes

**Claims Explanation:**

This paper is on the topic of test-time domain adaptation (TTA) and in particular makes the important observation of a "over-certainty" phenomenon inherent in entropy-minimizing TTA algorithms (which are a popular family in TTA), that lead to over certainty in predictions. The author address this by proposing a method that dynamically regulates the certainty of pseudo-labels using a temperature that is determined both by the entropy and the logit norm. The authors provide extensive experiments across multiple datasets and architectures, demonstrating that DEC improves calibration metrics (ECE, NLL) while maintaining competitive accuracy with state-of-the-art TTA methods.

The authors claim that the proposed method improves calibration while maintaining competitive accuracy. All 3 reviewers agreed that this claim is backed by sufficient evidence, through thorough experiments that consistently show strong results. Reviewer iKRA: "Extensive experiments across domain shifts demonstrate state-of-the-art calibration performance (measured by ECE and NLL) while maintaining competitive accuracy." Reviewer F8Wi: "The results look promising, and DEC's algorithm can resolve the over-confidence/under-confidence problem in many datasets and tasks... the results show an impressive improvement.". Reviewer 56nD: "Thorough Experiments: Extensive empirical validation across diverse datasets and backbones; statistical significance is reported."

---

> ### Author Response · Authors · 2025-06-19
> **Thank you for the helpful feedback and interest!**
>
> The authors would like to thank the AE and the reviewers for the helpful feedback and discussion on our work! We look forward to submitting the camera-ready version.